# Removal Mechanism and Effective Current of Electrocoagulation for Treating Wastewater Containing Ni(II), Cu(II), and Cr(VI)

**Chien-Hung Huang** [1], **Shan-Yi Shen** [2], **Cheng-Di Dong** [3], **Mohanraj Kumar** [2] and **Jih-Hsing Chang** [2,*]

[1] Department of Health Wellness and Marketing, Kainan University, Taoyuan 33857, Taiwan; huang@mail.knu.edu.tw

[2] Department of Environmental Engineering and Management, Chaoyang University of Technology, 168, Jifeng E. Rd., Taichung 41349, Taiwan; shanyi0226@gmail.com (S.-Y.S.); mohan1991mpt@gmail.com (M.K.)

[3] Department of Marine Environmental Engineering, National Kaohsiung University of Science and Technology, Kaohsiung 81157, Taiwan; cddong@nkust.edu.tw

[*] Correspondence: changjh@cyut.edu.tw

**Abstract:** This study aims to clarify the removal mechanism and to calculate the effective current of electrocoagulation (i.e., EC) for treating wastewater containing Ni(II), Cu(II), and Cr(VI). The adsorption behavior of various heavy metals onto $Al(OH)_3$ coagulant generated by the EC process was investigated and the estimating method of the corresponding current was established. Results indicate that adsorption of single Ni(II) and Cu(II) by $Al(OH)_3$ coagulant can be simulated by the Langmuir isotherm, while Cr(VI) adsorption fits the Freundlich isotherm better. As treating single heavy metal of wastewater, the removal mechanism of the EC process is the adsorption reaction. Under the coexisting condition, the Ni(II) and Cu(II) will compete for the same active sites on the $Al(OH)_3$ surface and Cu(II) suppresses Ni(II) adsorption. As treating the coexisting heavy metals, Ni(II) removal not only associates with adsorption but also with the coprecipitation. In contrast, Cr(VI) does not compete with other metal ions for the same type of adsorption sites. Whether single or coexisting conditions, the adsorption capacity of heavy metals onto $Al(OH)_3$ coagulants can be used to compute the necessary current to effectively remove heavy metals in the EC system.

**Keywords:** adsorption; aluminum hydroxide; electrocoagulation; heavy-metal wastewater

## 1. Introduction

Wastewater from metal-finishing facilities is troublesome for bio-treatment due to its complexity and toxicity. A typically waste stream from metal-finishing facilities usually contains different kinds of metal ions such as copper, nickel, cadmium, and chromium ions, along with chelating agents, oil and grease, organic solvents, and suspended solid particulates [1]. In addition to the complexity, the concentration of these heavy metals in wastewater is so high that further biological processes fail. Two significant reviews in recent years have summarized traditional and emerging techniques that apply to treating such wastewater [2,3]. Among the methods reviewed are chemical precipitation, coagulation/flocculation, adsorption, electro-coagulation, flotation, ion exchange, and membrane filtration. Chemical precipitation, especially hydroxide precipitation, was adapted early and is still recommended by US EPA for metal ion removal [1,4]. Coagulation is employed for the removal of contaminants in suspended or colloidal forms. Adsorption is the binding mechanism between soluble contaminants and solids that can provide a significant amount of active surface.The electro-coagulation (i.e., EC)

integrates the chemical precipitation, coagulation, and adsorption. The EC technique is based on the electrochemical reaction at the anode under alkaline pH condition, which leads to the removal function of hydroxide precipitation, coagulation, and adsorption (i.e., soluble pollutants of wastewater are adsorbed on the coagulant). Due to its high removal efficiency and easy operation, the EC system has been employed as the pretreatment process for the metal-finishing industry in Taiwan.

In an EC system, one pair of electrodes is installed in the electrochemical reactor and the anode (e.g., iron or aluminum) is used to produce the coagulant. With aluminum as the sacrificial anode, aluminum ions ($Al^{3+}$) and the hydroxyl group ($OH^-$) are produced at the anode and cathode, respectively. The precipitation of aluminum hydroxides will present under the alkaline pH condition. Then, aluminum hydroxide precipitate acts as an adsorbent at pH between 5.5 and 8 to adsorb heavy metal ions and acts as a coagulant above pH 8 to promote the coagulation process. In past research, extensive work has been conducted to explore the potential of EC for industrial wastewater treatment [5–8]. Some efforts focused on the removal of heavy metal ions [9–11]. Huang et al. reported the effect of coexisting anions on the formation of aluminum coagulant, resulting in the removal of metal ions [11]. Another study was conducted for the US Air Force to evaluate the effectiveness of EC for the removal of chromium, cadmium, nickel, and fluoride from leachate generated in the processing of spent abrasive blast media [12]. Recently, several researchers have studied the removal efficiencies of multiple heavy metals from different wastewater by EC [13–18]. Through different operating parameters, the EC is capable of obtaining the satisfactory removal of heavy metals and the optimal operating conditions were documented. However, rare studies clarified the competitive adsorption of coexisting heavy-metal ions during the EC process.

Heidmann and Calmano proposed that the formation of heavy-metal hydroxides coupled with the coprecipitation with aluminum hydroxide is the main EC mechanism to remove zinc, copper, nickel, and silver ions [19]. However, Adhoum et al. reported that the optimal EC condition for heavy-metal removal was between pH 4 and 8 [20]. Accordingly, it needs reasonable theory to explain why the concentration of heavy-metal ions starts dropping as the pH is less than 7 during the EC process. In addition, the EC took 15 min to reduce Zn(II), Ni(II), and Cu(II) concentrations to acceptable levels, in contrast, the removal for Ag(I) and Cr(VI) was only 30% under 15-min treatment. In terms of removal kinetics, Adhoum et al. found that the removal of Cr(VI) was about five times slower than that of Cu(II) and Zn(II). Apparently, the adsorption behavior of various heavy metals in the EC system plays a key role in removal. Akbal and Camci reported that with aluminum sulfate coagulation, 99.9% removal of Cu(II), Ni(II), and Cr(VI) was achieved by dosages of 500, 1000, and 2000 mg/L, respectively. They proposed that the removal of these metal ions was due to the combination of coagulation, adsorption, and coprecipitation [21]. Although significant progress has been made, the quantitative adsorption relationship between heavy metal ions and the EC coagulant is still unclear, which results in the difficulty of applying the EC process.

The EC process is usually employed to treat industrial wastewater containing complex heavy-metal compounds and organic chemical agents [22–24]. Before the EC application, the adsorption capacity of the coagulant for removing target heavy metals should be calculated correctively. Based on this adsorption capacity, the operational parameter of EC electricity can be computed to produce the appropriate amount of coagulant. Consequently, the EC process can effectively achieve the removal efficiency. In order to construct the above quantitative association, the adsorption model of coexisting nickel, copper, and chromium in the EC process (aluminum as the sacrificed anode) was established to clarify the competitive adsorption behavior of the above three heavy metals. Moreover, this adsorption model with the EC process was applied to treat wastewater samples collected from an electroplating facility to verify its practicability.

## 2. Materials and Methods

### 2.1. Metal Removal of EC System

At the cathode of the EC system, local hydroxyl ions can be generated due to water and oxygen reductions as the following equations [25]:

$$2H_2O + 2e^- \rightarrow 2OH^- + H_2 \tag{1}$$

$$2H_2O + O_2 + 4e^- \rightarrow 4OH^- \tag{2}$$

With an aluminum sheet as the consumable anode, $Al^{3+}$ ions will be produced by the oxidation reaction.

$$Al \rightarrow Al^{3+} + 3e^- \tag{3}$$

In the bulk solution, aluminum hydroxide will precipitate out as colloidal suspended particles in an EC system.

$$Al^{3+} + 3OH^- \rightarrow Al(OH)_3 \tag{4}$$

$Al(OH)_3$ is one kind of amphoteric hydroxide, that is, most of $Al(OH)_3$ will be formed between pH 5 and 8 (the lowest solubility at pH 6.4). The existence of $Al(OH)_3$ serves as the coagulant to remove the heavy metal ions in the wastewater. As the EC operation prolongs, the solution pH will increase beyond 8 and $Al(OH)_3$ will gradually dissolve in the alkaline solution. Under the high pH status of the EC system (pH above 8), precipitates of metal hydroxides such as $Ni(OH)_2$, $Cu(OH)_2$, and $Cr(OH)_3$ are formed. In contrast to $Cr(OH)_3$, hexavalent chromium (Cr(VI)) ion does not react with $OH^-$ at a high pH condition. Its removal is most likely due to adsorption by the coagulant of $Al(OH)_3$. The other possible removal method of Cr(VI) in an EC process is that the Cr(VI) ion may be reduced to Cr(III) in the vicinity of the cathodes [19]. Then, the Cr(III) ions react with $OH^-$ at a high pH to precipitate in the wastewater.

$$Cr^{6+} + 3e^- \rightarrow Cr^{3+} \tag{5}$$

### 2.2. Adsorption Isotherms of Metal Ions

As described in the previous section, heavy-metal ions can be adsorbed on the surface of aluminum hydroxide at a pH below 8. In the industrial wastewater, the EC process (aluminum as the sacrificed anode) should remove the nickel, copper, and chromium ions simultaneously. Consequently, the quantitative association between Ni, Cu, Cr ions and aluminum hydroxide should be established to clarify competitive adsorption behaviors. In this study, Freundlich and Langmuir adsorption isotherms were deployed to fit experimental data, respectively. The Freundlich isotherm describes adsorption as the following equation:

$$q = Ky^n \tag{6}$$

where $q$ is the adsorption capacity of heavy metal ions on the adsorbent (mg/g), and y is the equilibrium concentration of heavy metal ions in the bulk solution (mg/L). The constants, $n$ and K, are constants that depend on the nature of the heavy-metal ions and the adsorbent at a particular temperature. These two constants can be determined by a log-log plot of q versus y.

The Langmuir isotherm has a strong theoretical basis. The basis relies on a postulated chemical reaction between solute and vacant sites on the absorbent surface. The model is expressed as:

$$q = q_o \, K \, y/(1 + K \, y) \tag{7}$$

where $q_o$ is the adsorption capacity of heavy metal ions on the adsorbent (mg/g). The way to obtain the value of $q_o$ and $K$ is to plot $(q^{-1})$ versus $(y^{-1})$, as the following equation, the intercept is at $(q_o^{-1})$ and the slope is $(q_oK^{-1})$.

$$1/q = 1/q_o + 1/q_oK(1/y) \qquad (8)$$

### 2.3. Adsorption Experiments in an EC System

A bench-scale electro-coagulation system, as shown in Figure 1, was constructed to study the adsorption features of heavy-metal ions onto the $Al(OH)_3$ and the effectiveness of EC treatment for heavy metal removal. In addition, the schematic of the surface adsorption of metal ions in the electro-coagulation system is shown in Figure 2. The details of the chemicals information and the specification of the EC system can be found in a previous report [11]. Two series of experiments, i.e., indirect and direct EC tests, were designed to respectively distinguish the contribution of adsorption and coprecipitation mechanism on the removal of heavy metal ions from wastewater. For the indirect EC test, $Al(OH)_3$ was generated in the EC reactor and transferred to a test tube where a certain concentration of Ni, Cu, and Cr ions was prepared to react with the coagulant (i.e., $Al(OH)_3$ adsorption reaction). The pH was controlled in a tight range from 5.0 to 7.0 during whole the electro-coagulation process, to avoid the occurrence of hydroxide precipitation. The sulfuric acid (>95% of purity) and sodium hydroxide (>97% of purity) were used for acidic or basic adjustment of the solution. Four groups of test tubes were prepared; each group contained different concentrations of single Ni, single Cu, single Cr, and mixed Ni, Cu, and Cr ions, respectively. After a 3-min vigorous mixing and 1 h of settling, the pH and heavy-metal concentration of the supernatant was determined. The heavy metal adsorption capacity of the coagulant was calculated based on the heavy-metal concentration of supernatant. At last, the adsorption relationship between heavy-metal ions and the coagulant was quantified by Freundlich and Langmuir adsorption isotherms, respectively.

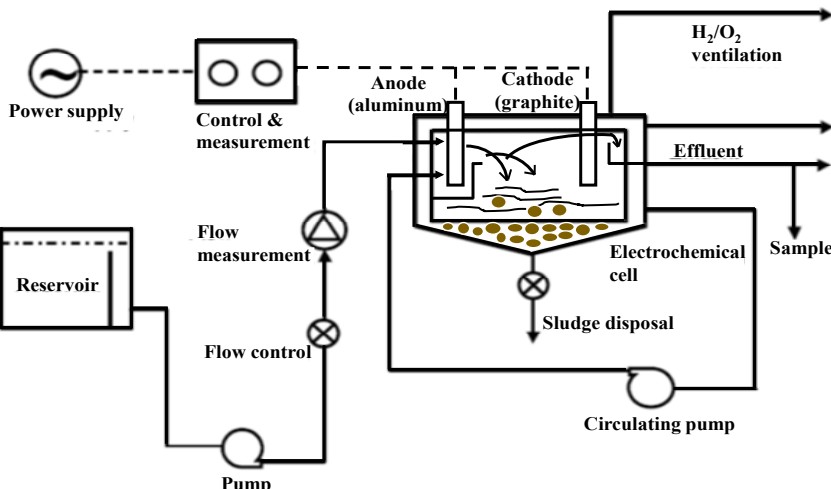

**Figure 1.** Flow diagram of the electro-coagulation test system.

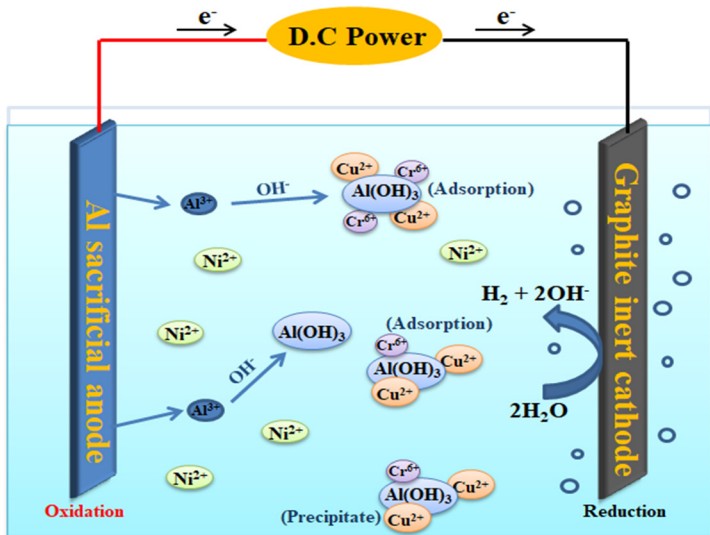

**Figure 2.** The schematic of the surface adsorption of metal ions in the electro-coagulation (EC) system.

For the direct EC test, the system was operated at a constant current of 1 ampere for 10 min. Solutions of Ni(II), Cu(II), and Cr(VI), with a 1.75 g/L of sodium chloride (NaCl) as the supporting electrolyte, were prepared and treated in the EC process. Once the Al(OH)$_3$ is generated in the EC reactor it would react immediately with heavy metal ions in wastewater. Treated wastewater of 10 mL was taken out every minute and let to settle for 1 h, then, the pH and heavy-metal concentration of the supernatant was determined. The same volume of fresh wastewater was recharged into the EC reactor after each sampling to compensate for the volume loss. In addition to the synthetic wastewater, the wastewater collected from an electroplating factory was treated by the EC process as well. The concentrations of Ni, Cu, Cr in all wastewater sample were determined by an IRIS Intrepid II inductively coupled plasma optical emission spectrometer (ICP-OES) produced by Thermo Electron Corporation, Franklin, MA. Experimental details of the above analysis are described in the Standard Methods for the Examination of Water and Wastewater [26].

## 3. Results and Discussion

### 3.1. Adsorption of Ni(II), Cu(II), and Cr(VI)

Figure 3 shows equilibrium concentrations of Ni(II), Cu(II), and Cr(VI) in the wastewater under the indirect EC procedure at different pH conditions. The dashed lines represent the solubility of Ni(OH)$_2$ and Cu(OH)$_2$ without the presence of Al(OH)$_3$ according to the USEPA report [4]. It can be noticed that the dashed line is almost parallel to the solid line of Ni and Cu, respectively. At the same pH, the Cu(II) concentration under the indirect EC procedure was about one order of magnitude lower than the solubility of Cu(OH)$_2$. The discrepancy could be attributed to the copper ions being adsorbed onto the generated Al(OH)$_3$ in the EC system. Such disparity about nickel hydroxide was even more significant, which was about three orders of magnitude. Heidmann and Calmano presented that the concentration of metal ions during an EC process declined as the pH was below 7 [19]. Apparently, the hydroxide precipitation cannot dominate the decrease of metal ions in a slightly acidic solution. However, most of Al(OH)$_3$ can be formed between pH 5 and 8. Hence, the coagulant adsorption is the major mechanism to remove heavy metal ions in the EC system, which agrees with the literature research [20,27,28]. Since hexavalent chromium cannot precipitate with the hydroxyl group, the decrease of Cr(VI) is simply due to its adsorption on the aluminum hydroxide.

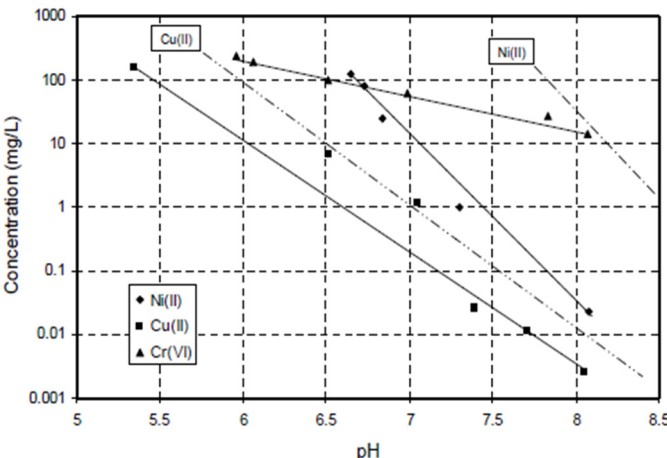

**Figure 3.** Equilibrium concentrations of Ni(II), Cu(II), and Cr(VI) in the wastewater under the indirect EC procedure at different pH conditions.

Figure 4a shows the adsorption curve of individual Ni(II), Cu(II), and Cr(VI) onto aluminum hydroxide at the status of pH 7. The abscissa is the metal concentration in the supernatant and the ordinate represents the metal ions adsorbed on the $Al(OH)_3$. Using Equation (6) can transform Figure 4a to a log-log plot as shown in Figure 4b, which deploys to gain adsorption parameters of Freundlich isotherm. It can be seen that the correlation coefficients are 0.82 and 0.86 for Ni(II) and Cu(II) adsorption, respectively. Experimental data of Figure 4a were fitted by Langmuir isotherm as well, whose linearization of Langmuir isotherm for Ni(II) and Cu(II) adsorption is presented in Figure 4c. Since correlation coefficients for Ni(II) and Cu(II) adsorption were above 0.9, Langmuir isotherm could describe the adsorption behavior of Ni(II) and Cu(II) ions onto the $Al(OH)_3$ coagulant. According to the intercepts of 0.003 and 0.0027 from Figure 4c, the adsorption capacity of Ni(II) and Cu(II) can be computed as 333 and 370 mg/g, respectively. In contrast to Ni(II) and Cu(II), the individual Cr(VI) adsorption onto the $Al(OH)_3$ coagulant at the status of pH 7 could be suitable to Freundlich isotherm (the correlation coefficient was 0.9868) as shown in Figure 4d. The different adsorption behavior from Ni(II) and Cu(II) may be attributed to Cr(VI) in an anion form of chromic acid (i.e., $Cr_2O_7^{2-}$). The adsorption capacity of Cr(VI) onto the $Al(OH)_3$ coagulant was estimated at around 200 mg/g. The adsorption capacity of each heavy metal is quite useful for environmental engineers to run the EC process. If we try to remove 100 mg Ni ions from the wastewater, for instance, the EC system should generate around 100 mg/333 mg/g = 0.3 g aluminum hydroxide. Since the molecular weight of $Al(OH)_3$ is 78 (atomic weight of aluminum is 27), the aluminum anode of the EC system should release (0.3 g/78) × 27 = 0.1 g aluminum. Based on Equation (5), we can obtain the electricity consumption of 0.011 mole electrons (i.e., (0.1/27) × 3 = 0.011). Consequently, the necessary current can be gained to control the EC system effectively.

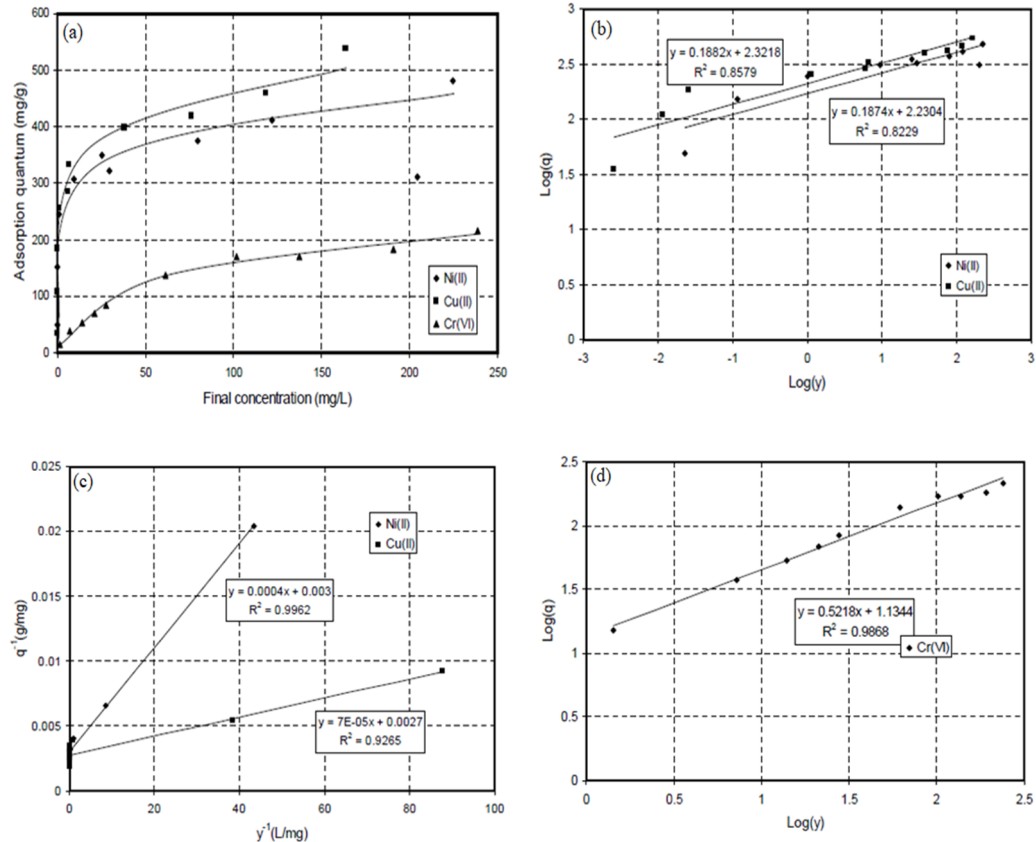

**Figure 4.** The adsorption curve and isotherm of different metal ions. (**a**) Adsorption curve of individual Ni(II), Cu(II), and Cr(VI) onto aluminum hydroxide at the status of pH 7. (**b**) Linearization of Freundlich isotherm for individual Ni(II) and Cu(II) adsorption. (**c**) Linearization of Langmuir isotherm for Ni(II) and Cu(II) adsorption. (**d**) Linearization of Freundlich isotherm for individual Cr(VI) adsorption.

### 3.2. Competitive Adsorption

Although the adsorption capacity of each heavy metal onto the aluminum hydroxide can be quantified by the above adsorption models, the complicated adsorption situation of coexisting heavy metals should be clarified as well. Figure 5a shows the competitive adsorption between Ni(II) and Cu(II) onto $Al(OH)_3$ at the status of pH 7. It could be observed that the Cu(II) adsorption behavior under the coexisting condition was similar to that under individual condition (shown in Figure 4a). However, a collapsed adsorption of Ni(II) occurred. Figure 5b shows the linearization of Langmuir isotherm for Ni(II) and Cu(II) competitive adsorption. With the competition between the two ions, Cu(II) adsorption was still consistent with the Langmuir isotherm, with a correlation coefficient of 0.95. However, there was no linear relationship that could be established between $(qo^{-1})$ and the equilibrium concentration of Ni(II) in the solution. In addition, the adsorption capacity of Ni(II) and Cu(II) was around 50 and 450 mg/g, respectively (shown in Figure 5a). The above phenomena indicate that the Cu(II) affinity to $Al(OH)_3$ is much greater than Ni(II).

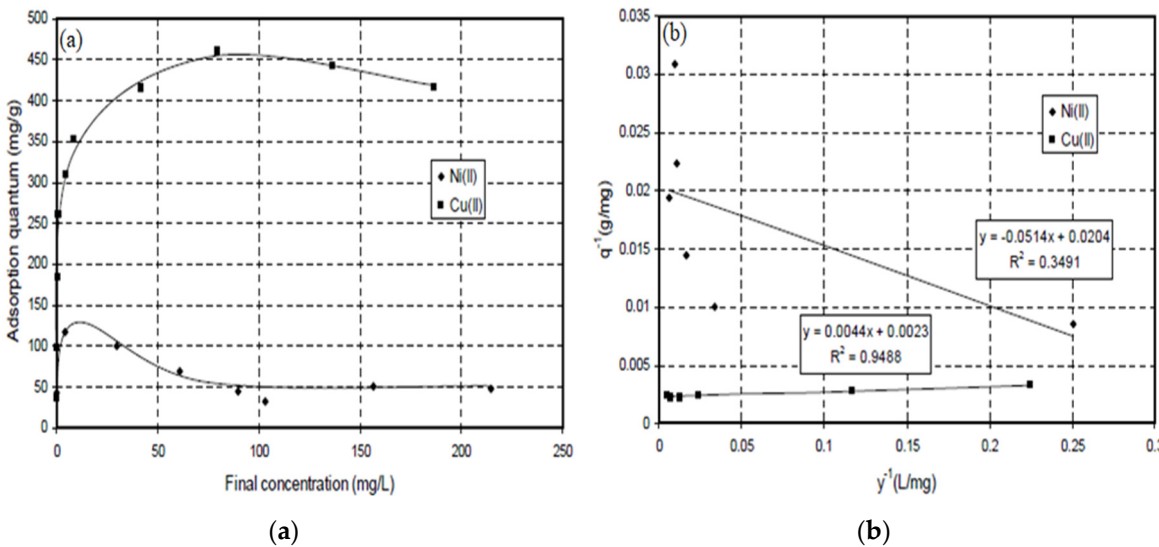

**Figure 5.** The competitive adsorption and isotherm linearization between Ni(II) and Cu(II) onto aluminum hydroxide. (**a**) The competitive adsorption between Ni(II) and Cu(II) onto aluminum hydroxide at the status of pH 7. (**b**) Linearization of Langmuir isotherm for Ni(II) and Cu(II) competitive adsorption.

In Figure 5a, it could be observed that the adsorption quantum of Ni(II) increased initially and declined gradually. Since vacant sites on the adsorbent (Al(OH)$_3$) surface are limited, the coexisting metals will compete to occupy the limited sites. When the concentrations of both Cu(II) and Ni(II) were relatively low, there were plenty of vacant sites available for adsorption of both Cu(II) and Ni(II). When the Cu(II) concentration increased dramatically, Cu(II) ions dominated the adsorption process (i.e., occupy most of the vacant sites and even replaced the Ni(II)). Kang and colleagues studied the adsorption behavior of multiple ions on Amberlite IRN-77 cation exchange resins and reported that Cr(III) could replace Cu(II) and Ni(II) ions adsorbed onto the resin [29]. Such replacement demonstrated that Cu(II) not only has a stronger adsorption affinity than Ni(II) onto aluminum hydroxide but also competes for the same sites on the aluminum hydroxide.

Figure 6a shows the competitive adsorption among Ni(II), Cu(II), and Cr(VI) onto aluminum hydroxide at the status of pH 7. In Figure 6a, Ni(II) and Cu(II) adsorption in the presence of Cr(VI) shows similar trends to those in Figure 5a. It is noticed that the adsorption capacity of Ni(II) and Cu(II) declined from 50 to 20 mg/g and from 450 to 350 mg/g, respectively. According to the illustration of Section 3.1, the adsorption sites of Al(OH)$_3$ for Cr(VI) were different from those for Ni(II), Cu(II). This implies that Cr(VI) does not compete with Ni(II) and Cu(II) for the same types of sites. Instead, it interferes with the adsorption process through other mechanisms. In addition, the adsorption capacity of Cr(VI) declined from 200 (individual condition) to 100 mg/g (competitive condition), Figure 6b shows the linearization of Langmuir isotherm for Ni(II) and Cu(II), and Cr(VI) competitive adsorption. In the presence of Ni(II) and Cu(II), Cr(VI) adsorption fits Langmuir well rather than the Freundlich isotherm, with a correlation coefficient of 0.99. The above results also demonstrate the interference mechanism among coexisting ions. Based on the adsorption capacity of each heavy metal under coexisting condition, like the example in Section 3.1, the necessary current can be obtained to control the EC system effectively.

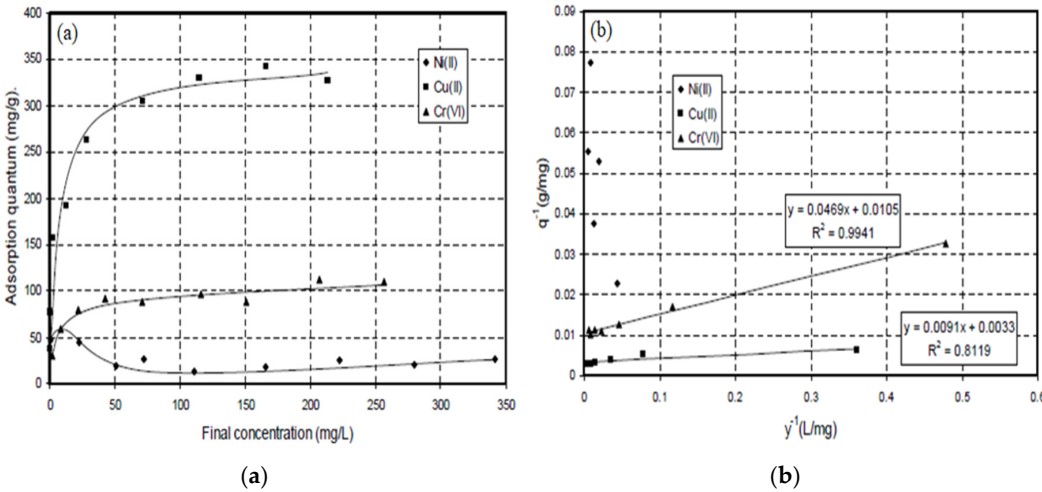

**Figure 6.** The competitive adsorption and isotherm linearization among Ni(II), Cu(II), and Cr(VI) onto aluminum hydroxide. (**a**) The competitive adsorption among Ni(II), Cu(II), and Cr(VI) onto aluminum hydroxide at the status of pH 7. (**b**) Linearization of Langmuir isotherm for Ni(II) and Cu(II), and Cr(VI) competitive adsorption.

*3.3. Direct EC Treatment*

Figure 7 shows the removal of individual Ni(II), Cu(II), and Cr(VI) treated by a 10-min EC process. It could be seen that Ni(II) and Cu(II) was eliminated completely after 4–5 min treatment, however, Cr(VI) was decreased from 188.8 to 128.0 mg/L. Since the volume of wastewater was 200 cm$^3$ in the EC reactor and the initial concentration of Ni(II), Cu(II), and Cr(VI) was 119.9, 96.0, and 188.8 mg/L, respectively, the total amount of Ni(II), Cu(II), and Cr(VI) was around 24.0, 19.2, and 37.8 mg, respectively. The adsorption capacity of Ni(II), Cu(II), and Cr(VI) was obtained from Section 3.1 as 333, 370, and 200 mg/g, respectively. Under 1.0 Ampere and 10 min EC operation, the total amount of generated Al(OH)$_3$ was around 0.16 g (i.e., 1.0 A × 60 × 10 = 600 coulombs, 600/96,500 = 0.0062 mole electrons, (0.0062/3) × 78 = 0.16 g of aluminum hydroxide). This EC operation could theoretically remove 53.3 mg Ni(II), 59.2 mg Cu(II), and 32.0 mg Cr(VI), individually (i.e., 0.16 × 333 = 53.3 mg for Ni(II), 0.16 × 370 = 59.2 mg for Cu(II), and 0.16 × 200 = 32.0 mg for Cr(VI)). Apparently, the theoretical removed Ni(II) was much greater than the initial Ni(II) content in the wastewater (i.e., 53.3 mg > 24.0 mg). It is noticed that the initial amount of Ni(II) was around half of the removed ability of Al(OH)$_3$. Accordingly, a 5-min EC operation could adsorb 24.0 mg Ni(II) as shown in Figure 7. Likewise, the theoretical removed Cu(II) was much greater than the initial Cu(II) content in the wastewater (i.e., 59.2 mg > 19.2 mg). A 4-min EC operation could adsorb 19.2 mg Cu(II) completely. The above derivation can demonstrate that the adsorption capacity of heavy metals is beneficial to design and control the EC operation.

According to Figure 7, the Cr(VI) concentration was reduced from 188.8 to 127.9 mg/L for the 10-min EC treatment. The theoretical removed Cr(VI) was less than the initial Cr(VI) content in the wastewater (i.e., 32.0 mg < 37.8 mg). As a result, the Cr(VI) content could not be removed completely under such EC operation. Based on the adsorption capacity of Cr(VI), the removal efficiency of Cr(VI) should be around 85% (32.0/37.8 = 0.846). However, the Cr(VI) content removed only one-third of the initial content in the wastewater. This implies that some factors about Cr(VI) adsorption on the aluminum hydroxide during the indirect and direct procedures have not been clarified. Heidmann and Calmano reported a 50% removal of Cr(VI) in a 50 min treatment and proposed Cr(VI) reduced to Cr(III) at the cathode before precipitating as chromic hydroxide [19]. Yang and Kravets have reported that using a steel sheet as an anode, the EC process is very useful in removing Cr(VI) from a synthetic wastewater sample [9]. They confirmed that Cr(VI) was reduced to Cr(III) by ferrous ions (Fe$^{2+}$) produced in the process. The adsorption of reduced Cr(III) needs to be quantified, which may interpret the removal of Cr(VI) in the EC system.

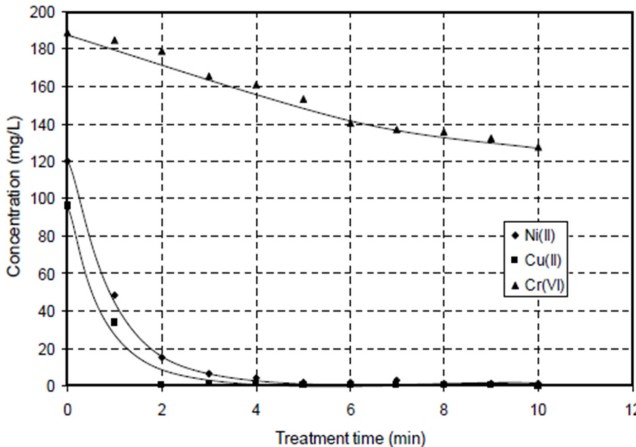

**Figure 7.** The removal of individual Ni(II), Cu(II), and Cr(VI) treated by a 10-min EC process.

For the actual condition of wastewater, regardless of the aluminum sheet or iron sheet used for electro-coagulation operation, various pollutants in wastewater can be removed through adsorption, coprecipitation, oxidation, and reduction mechanisms [30]. This study is focused on exploring the relationship between the removal mechanism of metal ions and competitive adsorption. The removal efficiency of various pollutants in wastewater by different electrodes has been reported from Tahreen et al. [31].

Figure 8 shows the removal of coexisting Ni(II) and Cu(II) throughout a 10-min EC process. During the treatment, the pH was kept between 4.8 and 6.0 to ensure that hydroxide precipitation was minimized. The 10 min of treatment could reduce Cu(II) concentration from 96.0 to 1.9 mg/L, which was similar to that without the competition from Ni(II) as shown in Figure 7. In comparison with the initial Cu(II) content and the adsorption capacity of Al(OH)$_3$, this removal phenomenon can be estimated as the same as the previous discussion. To consider the Ni(II) removal, the EC process reduced Ni(II) concentration from 111.3 to 12.6 mg/L. Such high removal efficiency was beyond our expectation since the adsorption capacity of Ni(II) was merely 50 mg/g. If the removal mechanism solely depends on the adsorption by Al(OH)$_3$, the final concentration of Ni(II) should be around 90 mg/L rather than 12.6 mg/L. This high removal efficiency of Ni(II) may be attributed to the coprecipitation mechanism [18,19].

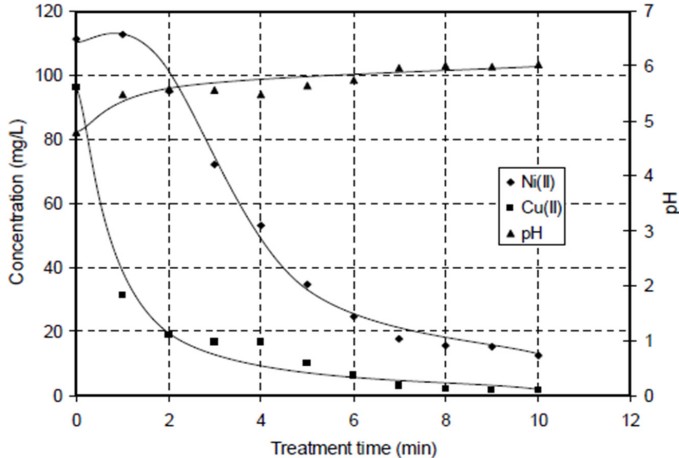

**Figure 8.** The removal of coexisting Ni(II) and Cu(II) throughout a 10-min EC process.

Figure 9 shows the removal of coexisting Ni(II), Cu(II), and Cr(VI) throughout a 10-min EC process. Basically speaking, the removal curves of Ni(II) and Cu(II) in Figure 9 are similar to those in Figure 8. This indicates that the presence of Cr(VI) influences the removal of Ni(II) and Cu(II) slightly. It could be noticed that the Ni(II) concentration remained unchanged at the early stage of the EC

procedure (shown in Figures 8 and 9), then, it dropped down to 26.6 mg/L at the end of the treatment. In contrast, the Cu(II) concentration dropped from 95.8 to 20 mg/L at the early stage. These different removal kinetics of Ni(II) and Cu(II) can be attributed to the fact that the Cu(II) ions dominated the adsorption process onto $Al(OH)_3$ (discussion in Section 3.1). As stated in a previous section, Cr(VI) does not compete with Ni(II) and Cu(II) to occupy the same type sites on the $Al(OH)_3$ surface. Instead, it may spatially or electrically block the sites or access to the sites. In Figure 9, the Cr(VI) concentration was reduced from 187.1 to 123.9 mg/L, which is almost identical to the EC treatment of single Cr(VI). This result was consistent with the hypothesis of Cr(VI) adsorption onto $Al(OH)_3$.

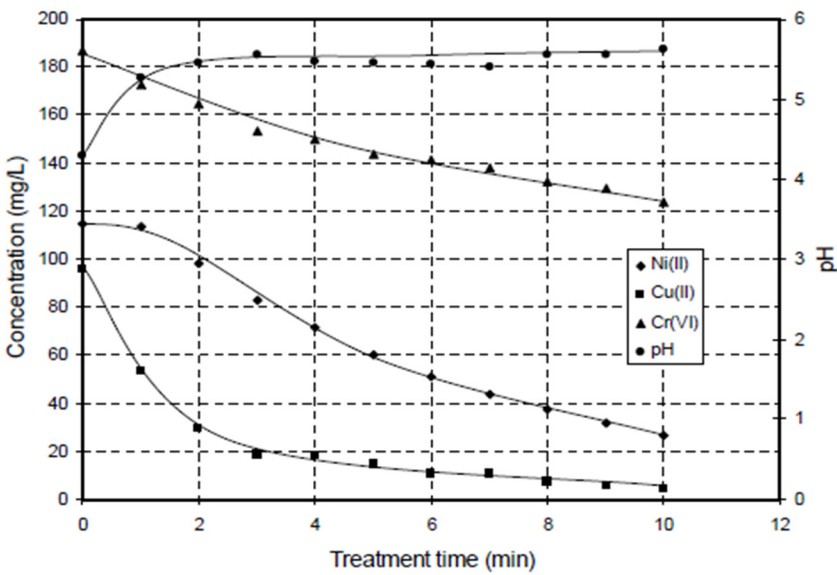

**Figure 9.** The removal of coexisting Ni(II), Cu(II), and Cr(VI) throughout a 10-min EC process.

### 3.4. Treatment of Electroplating Wastewater

To fully understand the interaction among the metal ions in the EC process, it would be beneficial to test the theory on a real sample. A wastewater sample was collected from an electroplating facility, which is based on various processes, including hard chromium, black nickel, bronze, and brass plating. The initial pH was 2.7, and the concentration of Ni(II), Cu(II), and Cr(VI) were 7.6, 4.7, and 280.9 mg/L, respectively. The sample was treated using the electrochemical process with aluminum as anode and graphite (as the inert electrode and has excellent conductivity) as the cathode, under a constant current of 1.5 amperes, and with 1.75 g/L NaCl as the supporting electrolyte. Figure 10 shows the concentration change of Ni(II), Cu(II), and Cr(VI) as a function of treatment time. During the 10 min of treatment, the solution pH increased steadily from 2.7 to 6.0. The Cu(II) and Cr(VI) ions in the sample behaved like those in the synthetic wastewater, as described in a previous section. Cu(II) concentration dropped 88% in 2 min, while 44% Cr(VI) was removed in the 10 min treatment. The dashed line in Figure 10 represents the change of Ni(II) concentration in synthetic wastewater from a previous section. When compared with the data from the treatment for real wastewater, it shows that the competition from Cu(II) was diminished significantly, while the interference from Cr(VI) was enhanced considerably. The above results demonstrate that the adsorption capacity of heavy metals plays a key role in the design and control of the EC operation.

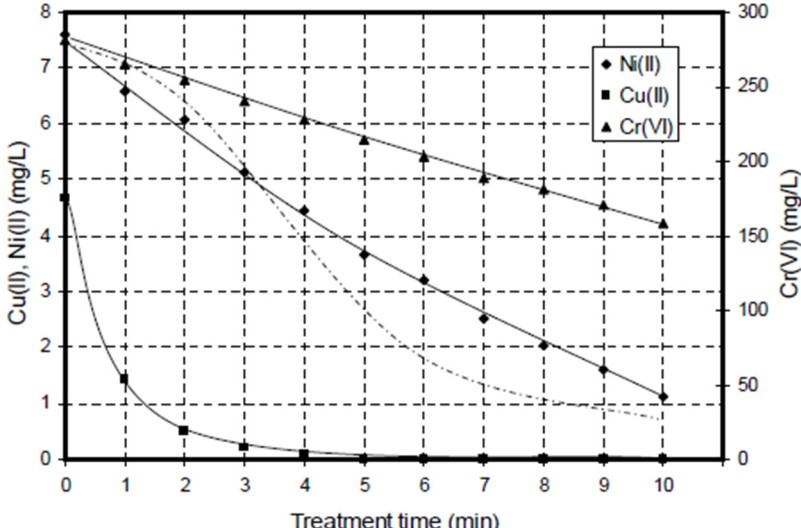

**Figure 10.** The EC treatment of wastewater sample from an electroplating facility.

## 4. Conclusions

Based on experimental results and discussion, several conclusions can be drawn as the following.

1. For individual ions, adsorption of Ni(II) and Cu(II) by Al(OH)$_3$ coagulant can be described by the Langmuir isotherm, while Cr(VI) adsorption fits the Freundlich isotherm better.
2. Treating a single heavy metal of wastewater, the removal mechanism of the EC process is the adsorption reaction.
3. Under the coexisting condition, the Ni(II) and Cu(II) will compete for the same active sites on the Al(OH)$_3$ surface and Cu(II) suppresses Ni(II) adsorption.
4. Treating the coexisting heavy metals, Ni(II) removal not only associates with adsorption but also with the coprecipitation. In contrast, Cr(VI) does not compete with other metal ions for the same type of adsorption sites.
5. Whether single or coexisting conditions, the adsorption capacity of heavy metals onto Al(OH)$_3$ coagulants can be used to compute the necessary current to effectively remove heavy metals in the EC system.

**Author Contributions:** Conceptualization and methodology: J.-H.C., C.-D.D. and C.-H.H.; formal analysis, investigation and data curation: C.-H.H.; validation, J.-H.C.; writing—original draft preparation: C.-H.H. and S.-Y.S.; writing—review and editing: M.K., and S.-Y.S.; supervision: J.-H.C. and C.-D.D. All authors have read and agreed to the published version of the manuscript.

**Funding:** This research received no external funding.

**Conflicts of Interest:** The authors declare no conflict of interest.

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
