# Peer review of "Removal Mechanism and Effective Current of Electrocoagulation for Treating Wastewater Containing Ni(II), Cu(II), and Cr(VI)"

_water, doi:10.3390/w12092614_

Round 1
Reviewer 1 Report
Although the idea of this paper is god, it was very difficult to understand the key details due to the poor English language. Thus, I recommend consulting an Editor to improve the language and resubmit it again.
Personally, I did not check the full paper due to this issue. I am happy to check the revised version.
Author Response
Dear reviewer,
Herewith I have attached the comment response.
Thank you.

Reviewer 2 Report
This manuscript describes the electrocoagulation (EC)-based heavy metal removal mechanism from wastewater. Authors used the various heavy metal (i.e., Ni(II), Cu(II) and Cr(VI)) to confirm the adsorption behavior, and different isotherms were applied to optimize the condition of adsorption reaction. While Languir isotherm could be used to promote the adsorption of Ni(II) and Cu(II), Freundlich isotherm could be used to promote the adsorption of Cr(VI). The reviewer thinks that the different isotherm-based removal mechanism of heavy metals might be used to optimize the adsorption of heavy metal from wastewater; however, this manuscript is still needed to revised to be more suitable for its publication. There are some comments. 1) Please revise the superscript and subscript errors of chemical formula in the manuscript. 2) Please check the typing errors in 106 lines EO→EC, in 252 lines Co→Cu, and in lines 43 add the ‘.’ the next of adsorption. 3) Please add the other reference to demonstrate the suggested mechanism. 4) It is recommended to provide the specific schematic of adsorption surfaces in the figure 1. 5) Please explain the reason why the authors choose the graphite as cathode. 6) Please discuss how to react the other materials in wastewater on the actual condition. 7) Please rearrange the figure set as this journal format.Author Response
Dear reviewer,
Herewith I have attached the comment response.
Thank you.

Round 2
Reviewer 1 Report
The authors has addressed the majority of the previously mentioned issues. However, the following issues must be addressed before further processing:
1- There are few grammar mistakes, such as (aims to (at not to)) it must be (aims at), (pollutants of wastewater is (are not is)). Check the rest of the paper please.
2- Title of section 2 can not be an adjective (2. Experimental), it must be changed into (2. Experimental work) or (2. Materials and Methods).
3- In Section 2 (experimental), you mentioned that (The pH was controlled in a tight range from 5.0 to 7.0, to avoid the occurrence of hydroxide precipitation). Do mean the initial pH value or it was continuous controlling process? Give more details please.
4- What chemicals did you used to have the heavy metal ions (Zn(II), Ni(II), Cu(II) Ag(I) and Cr(VI)).
5- All used equations must be referenced. The following references could be used for all EC equations:
- Hashim, K.S., AlKhaddar, R., Shaw, A., Kot, P., Al-Jumeily, D., Alwash, R. and Aljefery, M.H., 2020. Electrocoagulation as an eco-friendly River water treatment method. In Advances in Water Resources Engineering and Management (pp. 219-235). Springer, Singapore.
- Hashim, K.S., Al-Saati, N.H., Alquzweeni, S.S., Zubaidi, S.L., Kot, P., Kraidi, L., Hussein, A.H., Alkhaddar, R., Shaw, A. and Alwash, R., 2019, August. Decolourization of dye solutions by electrocoagulation: an investigation of the effect of operational parameters. In IOP Conference Series: Materials Science and Engineering (Vol. 584, No. 1, p. 012024). IOP Publishing.
- Hashim, K.S., Hussein, A.H., Zubaidi, S.L., Kot, P., Kraidi, L., Alkhaddar, R., Shaw, A. and Alwash, R., 2019, September. Effect of initial pH value on the removal of reactive black dye from water by electrocoagulation (EC) method. In Journal of Physics: Conference Series (Vol. 1294, No. 7, p. 072017). IOP Publishing.
Author Response
Dear Reviewer,
Herewith I have attached the comment document.
Thank you.

Reviewer 2 Report
.
Author Response

(The authors gave the same response as above.)
